# Position: Vision Encoders should be Image Size Agnostic and Task Driven

**Nedyalko Prisadnikov** [1]  **Danda Pani Paudel** [1]  **Yuqian Fu** [1]  **Luc Van Gool** [1]

## Abstract

This position paper argues that the next generation of vision encoders should be image size agnostic and task driven. The source of our inspiration is a behavioral trait of biological vision – *efficiency*. We focus on a couple of ways in which vision in nature is efficient, but modern vision encoders not. Humans and animals deal with vast quantities of visual data, and need to be smart where they focus their limited energy. It is our belief that vision encoders should be dynamic and the computational complexity should depend on the task at hand rather than the size of the image. To realize this, we introduce SOVA (**S**equential **O**bservation with **V**isual **A**ttention) – a *proof-of-concept* solution for image classification. Despite classification not being representative for what we are trying to achieve, it shows that our approach is feasible and promising. Code is available at: `https://github.com/insait-institute/sova`.

## 1. Introduction

With this position paper, we aim to spark a renewed biological inspiration for computer vision models, specifically vision encoders. Our source of inspiration is not structural, but behavioral: we are interested in the *efficiency* of biological vision.

Why efficiency? As noted by (LeCun, 2025), a four-year-old child has processed more bytes of visual data than the largest text corpora used to train modern large language models (LLMs). Reading that amount of text would take a human hundreds of thousands of years. This discrepancy highlights that the volume of visual data we encounter daily is immense. Consequently, biological systems must be highly

selective, spending processing energy only where it yields the highest utility (Lennie, 2003). A guiding principle of this work is that *the goal of vision is not to process and understand every detail of what we are seeing, but to extract biologically relevant information*.

While many human inventions are inspired by nature, the final designs are often shaped by engineering trade-offs. We built planes inspired by birds, yet planes do not flap their wings. Similarly, Convolutional Neural Networks (CNNs) (LeCun et al., 1998) draw inspiration from the visual cortex (Hubel & Wiesel, 1962) (e.g., weight sharing, hierarchical feature extraction) but rely on bottom-up computation that diverges from biological processing. More recently, with the advent of the transformer (Vaswani et al., 2017), the Vision Transformer (ViT) (Dosovitskiy et al., 2021) has become the dominant architecture, offering global receptive fields and the ability to quickly attend to any part of the image. However, standard ViTs are computationally inefficient, operating at a constant resolution with a quadratic dependence on token count (image size).

As we think that efficiency is a crucial aspect of vision in nature, we would like to inspire future research on vision encoders that are built from the ground with efficiency in mind. In our opinion a **path towards better and more efficient vision encoders, is one that focuses on models that are task-driven and image size agnostic**.

## 2. Why Image Size Agnostic?

A fundamental inefficiency shared by current vision models and LLMs is that the computational cost to solve a problem depends more on the *size of the input context* than on the *difficulty of the task*. We believe this dependency should be inverted.

Despite hardware advances, vision encoders have not kept pace with modern camera capabilities. Cameras now routinely capture tens or hundreds of millions of pixels, yet encoders often resize these down to small, fixed resolutions (e.g., $224 \times 224$) or suffer quadratic compute costs. Even a linear dependence on pixel count is inefficient when data volume is vast.

[1]INSAIT, Sofia University "St. Kliment Ohridski". Correspondence to: Nedyalko Prisadnikov <first.last@insait.ai>.

*Proceedings of the $43^{rd}$ International Conference on Machine Learning*, Seoul, South Korea. PMLR 306, 2026. Copyright 2026 by the author(s).

A core limitation is the *bottom-up* processing paradigm, which treats all pixels equally. This is true for both ViTs with their fixed sized patches, but also for CNNs applying the same filters throughout the area of the image. Image data being vast, it is also highly redundant and compressible. Compressible means that not all parts of the image bring equal value. Consider an image where the top half contains only cloudless sky. With a ViT half of the patches will contain pixels from the sky. Perhaps the amount of information this part of the image brings could be contained in a single patch, or even less. This is task dependent. If you are a painter that tries to draw the scene, maybe you are interested even in minute variations of color and tone.

Alternatively, biological vision approaches processing visual data via foveation. The fovea – a roughly 1–2° circle of high acuity around the center of gaze – allows the brain to sample the world at high resolution only where necessary. Outside this region, visual acuity declines quickly and significantly (Strasburger et al., 2011). To construct a detailed internal representation of our surroundings, we must continually shift our gaze (saccades) and refocus on successive regions of interest, a dynamic process fundamentally driven by the visual task at hand (Yarbus, 1967). We believe this contains valuable hints on how to make computer vision encoders image size agnostic. We have to extract features in a top-down manner at resolution determined by the system's *eye*, not by the size of the image. If the resolution of the image is lower than the resolution of our system, then we do not see enough detail. If the resolution is higher, then it gives us an ability to zoom. A major contribution of this work is a complete method for extracting patches in top-down manner, independently of the image size.

Additionally, this implies an iterative process. Instead of a single pass, the model performs multiple steps (gazes), each with a fixed context size. This decouples computational complexity from input image size. Section 7 introduces our method for top-down, size-agnostic patch extraction. Each subsequent gaze(step) allows us to enrich the model's internal representation of the scene. Moreover, this dynamic process of selecting the next gaze, is where the opportunity lies for a task-driven behavior.

## 3. Why Task Driven?

We argue that vision encoders should be *task-driven* rather than *task-agnostic*. Extracting relevant information from vast visual streams is essentially a compression problem, and optimal compression is dependent on the downstream objective. We cannot expect to classify an image and find Waldo [1] with the same computational pattern. The two tasks require vastly different approaches and computational resources.

*It is vital to clarify that "task-driven" does not imply training separate models for each task.* On the contrary, we propose a *single, general-purpose vision encoder capable of dynamic computation*. Just as an LLM changes its output based on a text prompt, a vision encoder should dynamically adjust its inference path – deciding where to look and at what resolution – based on the specific requirements of the task at hand. A task-agnostic encoder models a probability distribution $P(Z|I)$, where $Z$ are the output features, and $I$ is the image data. A task-driven encoder models a probability distribution $P(Z|I,T)$, where $T$ is a task prompt.

In our opinion, using *task agnostic encoders is not only a case of inefficiency, but a fundamental limitation on how we use vision models today*. Imagine playing a game where you are given an image for a second, then the image is hidden and you are asked questions about its content. What was the image of? Were there people? How many? What is the color of the clothes of the leftmost person? What about their shoes? This game is very difficult. Asked about a small detail after the picture is hidden would make one struggle a lot. It might be impossible if attention was not paid to this particular detail. But if the question came before looking at the image, it becomes trivial. Handling our daily lives this way seems unimaginable, yet this is what we expect from computer vision models. Typically, when we use a vision encoder in a system, say a vision language model (VLM), we run the input image through the encoder, extract features from it and use that features downstream. We always extract the same features, no matter what the downstream task is. This is a lot like playing the game above. What if we process the text prompt first and use it to guide us on how to look at the image?

## 4. Proposed Solution

In this section, we outline the architectural components of SOVA, an image size agnostic and task-driven vision encoder. Although this is a position paper, we provide experimental evidence to validate the viability of our proposal. Our experiments are performed on the ImageNet-1K (Deng et al., 2009) dataset. While ImageNet—with its centered objects – is not the ideal environment to demonstrate the full potential of active visual exploration, the results presented here serve as a *proof-of-concept*. They demonstrate that an iterative, foveated approach is feasible without scaling the computational complexity with the size of the images.

Our solution builds upon the transformer architecture (Vaswani et al., 2017). Broadly, the proposed encoder comprises three components: 1) a **Transformer**, which serves as the core visual processing unit; 2) a **Policy**, which

---

[1]"Where's Waldo?" is a children's puzzle book by Martin Handford, Little, Brown and Company, 1987. "Where's Waldo?" is a registered trademark of Candlewick Press.

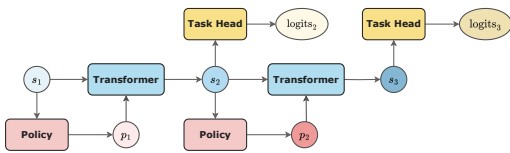

*Figure 1.* Overall system architecture, depicted as an **unrolled recurrent process** across multiple time steps. We evolve an initial state (prompt) $s_1$ by iteratively calling the same shared Transformer model with different patches. Crucially, the Policy, Transformer, and Task Head are single modules applied repeatedly, not separate networks per step. The patches $p_t$ to extract are selected by the policy based on the current state $s_t$. The task head produces a task-specific output (logits$_{t+1}$) given the updated internal state. We generate a task output after each iteration, allowing the sequence to scale to an arbitrary number of states until the task is solved.

determines the next gaze location based on the current understanding of the image; and 3) a **Task Head**, which decodes the internal state into task-relevant outputs (e.g., classification logits).

To accommodate high-resolution data with limited compute, the key is to use the transformer *iteratively*. At each step, the model processes only a small, selected set of image patches, evolving a recurrent internal state (or memory). The policy uses this state to decide where to look next, effectively "saccading" across the image. Finally, the task head reads the updated internal state to produce a prediction. It is worth noting that this computational paradigm is not entirely new; seminal works like (Schmidhuber & Huber, 1991; Mnih et al., 2014; Ba et al., 2014) explored similar directions. A major goal of this paper is to revive interest in these recurrent visual strategies, using modern architectural components.

Figure 1 provides a schematic overview of our framework, illustrated as an **unrolled sequence over time**. The Policy, Transformer, and Task Head share weights across all iterations; they are the same modules acting in a recurrent loop. The computation begins with a learned initial state $s_1$. In a **task-driven** setting, this initial state acts as a *task prompt*, conditioning the model's behavior from the very first step. At any step $t$, the policy takes the current state $s_t$ as input and selects the next set of patches $p_t$ to extract. The shared transformer ingests both the state and the new patches to produce an updated state $s_{t+1}$. At this stage, we can query the task head with $s_{t+1}$ to obtain a result and potentially terminate computation if confidence is high. Alternatively, the policy can use $s_{t+1}$ to select a new set of patches $p_{t+1}$. This cycle repeats, allowing the computational budget to scale dynamically as the task requires.

## 5. Vision-Language Models: A Call to Action

A very compelling application for task-driven, size-agnostic encoders are Vision-Language Models (VLMs). Current

state-of-the-art VLMs typically operate by pairing a powerful Large Language Model (LLM) with a vision encoder (e.g., CLIP (Radford et al., 2021) or SigLIP (Zhai et al., 2023)). In this paradigm, the visual encoder processes the image *independently* of the user's text prompt. Mathematically, the encoder models the distribution $P(Z|I)$, extracting a fixed set of visual tokens $Z$ from image $I$. The LLM then models $P(Y|Z, T)$, generating the response $Y$ conditioned on those pre-computed visual tokens and the text prompt $T$.

This brings us back to the game analogy from Section 3. By decoupling feature extraction from the prompt, we force the VLM to play the challenging game: the encoder must compress the entire image into $Z$ without knowing which details will be relevant later. If the prompt asks about a minute detail – e.g., "What is the time on the watch on the person?" – and the fixed encoder has already discarded that detail in favor of global semantic features, the LLM is unable to answer, no matter how powerful its reasoning capabilities are.

A task-driven encoder as proposed in this paper provides a fundamental shift in how vision encoders are integrated into VLMs. Instead of a static perception module, we can have a *text-guided feature extraction* process. The vision encoder models $P(Z|I, T)$ directly. This allows it to actively "look" at the image according to the task.

### 5.1. Architectural Integration

Integrating a dynamic iterative encoder into VLMs is a natual extension to the existing models. Architectures like BLIP-2 (Li et al., 2023) already employ a "Q-Former" to bridge the modality gap. The Q-Former uses learnable queries to extract features from a frozen vision encoder, conditioned on text. However, the Q-Former is limited to querying *already computed features*. It cannot revisit the raw image data to recover missing details. While the current models focus on *image-grounded text generation*, we propose a focus on *text-grounded image exploration*.

### 5.2. A Call to Action

While vision language aspects are beyond the technical scope of this paper, we would like to frame it as a call to action to the community. We believe that *text-grounded image exploration* would be an exciting approach to vision language models. Such approach would require developing training objectives that align the active exploration policy of the vision encoder with the reasoning capabilities of LLMs.

## 6. Iterative Transformer with Internal State

How can we use a transformer iteratively? How can it evolve an internal state? The most popular vision transformer

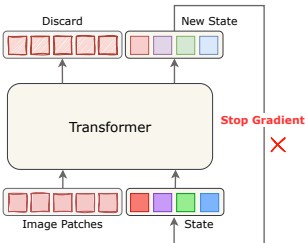

*Figure 2.* Schematic view of how we keep an evolving internal state with the transformer. It processes patch tokens together with the previous state as input. After multiple self-attention layers between the input we get updated state together with updates patches than are discarded. In the following iteration they will be replaced by different ones.

ViT (Dosovitskiy et al., 2021) is based on BERT (Devlin et al., 2019). There a sequence of input tokens (patches) and a learned CLS token get transformed using full self-attention between them. The CLS token can be treated as an internal state. Inspired by the success of the DETR (Carion et al., 2020) transformer decoder with $N$ learned object queries, we expand the internal state from a single CLS token to a set of $N$ learned tokens.

The transformer we use is basically the same as a ViT with registers (Darcet et al., 2023). However, instead of keeping the transformed patches and discarding the registers, we do the opposite. The input to the transformer is a set of patches and $N$ learned state vectors. After we run the transformer, we discard the transformed patches, and keep the evolved state, which we then use as input for the next iteration with different set of patches.

Figure 2 shows the application of the transformer. It is important to note that while the output state is input for the next iteration, we do not train the model as a recurrent neural network (RNN). We do not allow gradients to backpropagate between different steps. Our solution is similar to the Recurrent Memory Transformer (Bulatov et al., 2022). The main difference is that we do not perform backpropagation through time (Robinson & Fallside, 1987; Mozer, 2013; Werbos, 1988).

## 7. Extracting Patches in a Size Agnostic Way

**Bottom-up vs Top-down Approach.** How to extract patches is a critical component if we want to be image size agnostic. Most computer vision models work in a bottom-up fashion – they start from the pixels and build their way up. A ViT model splits the image in a grid of equal sized patches. For example, each patch is $16 \times 16$ pixels. This has two major consequences. First, poor scalability with the size of the image. The number of patches grows quadratically with the linear size of the image. Second, if we change the image size, we also introduce a distribution shift. What a

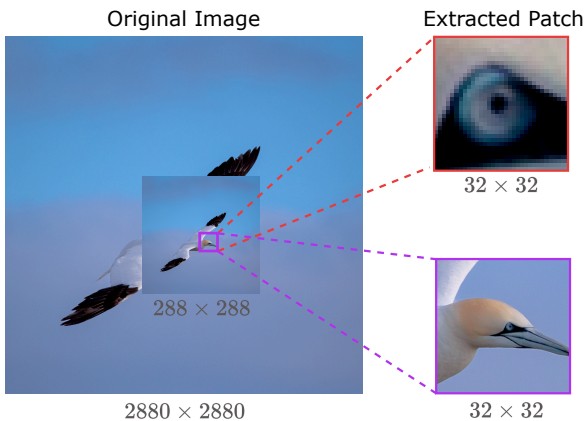

*Figure 3.* Example of distribution shift with ViT patches when changing the image size. The original size of this image is $2880 \times 2880$. On the right we see two patches (of size 32) extracted from a resized image ($288 \times 288$) and from the original image. You can see how much the type of visible detail changes from the resizing.

$16 \times 16$ pixels patch represent geometrically changes with the image resolution. Consider using the same image in two different sizes, $2000 \times 2000$ pixels, and $200 \times 200$ pixels. A patch in the smaller image will contain much larger portion of the image and will contain larger shapes and details. You can see the effect of this in Figure 3. Changing the image size significantly between training and testing can cause a significant distribution shift. This is why when training ViTs it is very important to use random resized crop augmentations (Touvron et al., 2019). This issue is not limited to ViTs. The same distribution shift happens for CNNs. This is because they extract features in a bottom-up fashion, starting from the pixels.

**Top-down Alternative.** We propose to use a top-down approach for extracting patches. A patch is not defined on the pixel level, but as proportion of the image size. That way changing the image size does not lead to change in the types of details a given patch sees. Additionally, we do not have to limit ourselves to patches of the same size. *In our proposal any square crop of the image is a valid patch.* Each square crop can be resized to a fixed size (in our case $16 \times 16$ pixels), and be fed to a tokenizer (or `PatchEmbed` module). This has a very useful property. If the square crop is small, then the change to the resized $16 \times 16$ version will be minimal, i.e. the patch is of high resolution, but covers small portion of the image. If the crop is a large square, the change to the $16 \times 16$ version is significant – a lot of detail is lost – i.e. the patch is of low resolution, but covers large portion of the image. In principal, this is very similar to the difference between the foveal (high acuity, small coverage) and peripheral (low acuity, high coverage) vision.

**Multi-Zoom Patches.** The definition above gives us a very flexible notion of what a patch is. However, in order to

## Multi-zoom patches

## Patches

*Figure 4.* Extracting multi-zoom patches – patches with the same center but varying sizes. Notice that the area of the crop outside the image is padded with 0s.

be computationally efficient we want to extract patches from images in a systematic way which is scalable on modern highly parallel computer architectures. As mentioned a patch can be any square crop. Let us say the size of the crop is $C \times C$. We define a patch using three coordinates $(x, y, z)$. $(x, y) \in [0, 1]^2$ are the coordinates of the center of the crop. This is in relative coordinates, i.e. both values are between 0 and 1. $z$ is a non-negative number that represents the zoom level of the crop. If $z = 0$, then the size of the crop $C = min(H, W)$. When $z > 0$, then the size of the crop is $C = min(H, W)/2^z$. Here, $H$ and $W$ are the height and width of the image respectively. This is the essence of our top-down approach. The size of the patch is relative of the image size. If the image size changes a patch remains a crop of the same part of the image.

To make extracting patches more efficient and parallelizable on modern hardware, we always extract a fixed sequence of patches from a given center $(x, y)$. In other words given a patch center $(x, y)$ we extract $M$ patches, one for each value of $z = \text{jnp.linspace}(0, \max_z, M)$. This way, for a given gaze location $(x, y)$ we get a series of patches of decreasing acuity and increasing image coverage. See an example of multi-zoom patches in Figure 4. This is similar to how our eyes work. We see only a small circle around the center of gaze sharply. While our overall field of view is large it is of low definition. Figure 5 shows multi-zoom patches overlayed over each other. The patches are first resized to their original size from $16 \times 16$. This is why the largest patch is very blurry. Note that this is not what the transformer sees. It is only drawn for us to appreciate the similarity between

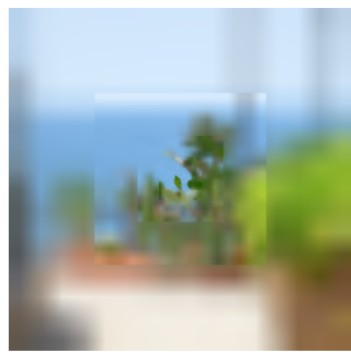

*Figure 5.* Example multi-zoom patches overlayed over each other after being resized to their original size. The center here is sharp due to the patch with highest zoom level. While the patch with $z = 0$ covers the whole image, but it is of very low acuity.

the multi-zoom patches and our own foveal-peripheral vision. The patches that the transformer sees are the ones in Figure 4. In Appendix A.1 we provide implementation for multi-zoom patch extraction in JAX. Our multi-zoom patches are similar to the ones used in (Mnih et al., 2014). They also extract series of patches from the same center each twice the dimensions of the previous. However, a crucial difference is that there the approach is bottom-up. We extract the patches in top-down manner and hence our approach is image size agnostic.

## 8. Learned Policy for Where to Look

The last remaining part is learning where to look. This is the most under-explored area of our proposal and the key to scaling our idea to many tasks. In this prototyping stage, we use reinforcement learning through gradient based policy optimization (Williams, 1992). More specifically we use the Group Relative Policy Optimization (GRPO) (Shao et al., 2024) algorithm. With the multi-zoom patches we only need to specify the gaze location in order to extract patches for the next step through the transformer. Thus, we can model our policy with continuous actions in the box $[0, 1]^2$.

With this approach one of main challenges is how to train the transformer and the policy end-to-end. The internal state of the transformer is the visible observation for the policy. If it changes during training it is very difficult to keep the policy relevant. Additionally, the training loops and dynamics for training the transformer and the policy using policy optimization are quite different. At this proof-of-concept stage we train the system in two stages. First, we pretrain the transformer and the task head using random policy, i.e. at each iteration we choose a gaze location uniformly at random. In the second stage we freeze the

transformer and optimize the policy using GRPO. More details on our approach can be found in Appendix A.2.

## 9. Proof-of-Concept Results

We use ImageNet-1K (Deng et al., 2009) to benchmark the feasibility of SOVA. While standard image classification is not representative of what we are trying to achieve, it provides an established environment to validate the core components of our system. We omit smaller benchmarks like MNIST (LeCun et al., 1998) or CIFAR (Krizhevsky et al., 2009), because with very small images it is hard to test the image size agnostic part of the encoder. All patches will contain significant portion of the image.

Full implementation details and hyperparameters are provided in Appendix A.3. The primary objective of these experiments is to validate three hypotheses:

1. **Input Feasibility:** Multi-zoom patches are a suitable input representation for vision transformers.

2. **Memory Evolution:** An iterative transformer can successfully accumulate information over time, without requiring backpropagation through time.

3. **Active Perception:** It is possible to train a policy that learns *where* to look to maximize task performance.

### 9.1. Validating Memory: Iterative Transformer

First, we test the model's ability to maintain and evolve a memory state. As a baseline, we use a standard ViT-Base (patch size 16) trained on $256 \times 256$ images (256 patches in total). To test our iterative approach, we split these 256 patches into 4 random subsets of 64 patches. The model processes these subsets sequentially (in 4 steps), carrying over a hidden state of $N = 32$ vectors between the steps. Crucially, **no gradient backpropagation occurs between the steps**.

Table 1 compares this approach to the standard ViT. Despite the structural disadvantage – our model never performs global self-attention across the whole image – it recovers a significant portion of the baseline performance ($72\%$ vs $78\%$). This confirms that the internal state successfully functions as a working memory, aggregating features from disjointed observations into a coherent global representation. Notably, the validation accuracy increases monotonically with each step ($Step\ 1 < Step\ 2 < \dots$), demonstrating that the model actively accumulates information rather than forgetting it.

### 9.2. Validating Policy: GRPO with Multi-Zoom Patches

Next, we integrate the multi-zoom patches and a learned policy. In this setup, the model sees $M = 16$ multi-zoom

*Table 1.* Iterative training with shuffled groups of ViT patches.

| Method | Acc @ Top 1 | Acc @ Top 5 |
|---|---|---|
| Iterative (Step 1) | 0.66 | 0.86 |
| Iterative (Step 2) | 0.69 | 0.88 |
| Iterative (Step 3) | 0.71 | 0.89 |
| Iterative (Step 4) | **0.72** | **0.90** |
| ViT (Baseline) | 0.78 | 0.93 |

patches centered at a specific $(x, y)$ coordinate per step. This creates a challenging "keyhole" view of the world: the model never sees the full image at high resolution. We first pretrain the transformer together with the task head using a random policy, i.e. we train for $8$ steps, at each step selecting a gaze location uniformly at random. Then, we freeze the transformer and task head, and train the gaze policy using GRPO.

Table 2 presents the results. The performance gap compared to the full-context baseline is expected given the severe information bottleneck – the model views only a fraction of the image pixels compared to a standard ViT. However, a few important observations show the feasibility of our proposal:

1. **Handling of Multi-zoom Patches:** The `PatchEmbed` tokenizer module is shallow, but it is capable of handling patches of varying size and acuity.

2. **Effective State Dynamics:** Both the random and learned policies consistently improve with more steps. This indicates that the transformer successfully integrates the foveated glimpses into it memory state.

3. **The Value of Active Perception:** While the random policy achieves reasonable performance (due to the low-resolution context often being enough for classification), the learned policy outperforms it, particularly in the early steps.

*Table 2.* Performance of SOVA with a learned policy (GRPO) vs. random policy using multi-zoom patches.

| Model Variant | Acc@Top 1 | Acc@Top 5 |
|---|---|---|
| Rand Policy (Step 1) | 0.36 | 0.58 |
| Rand Policy (Step 4) | 0.55 | 0.78 |
| Rand Policy (Step 8) | 0.60 | 0.82 |
| Learned Policy (Step 1) | 0.51 | 0.74 |
| Learned Policy (Step 4) | 0.62 | 0.83 |
| **Learned Policy (Step 8)** | **0.65** | **0.85** |

While this performance is bounded by the difficulty of classifying complex scenes through narrow glimpses, these results serve as a strong proof-of-concept. They demonstrate that a policy-driven, iterative encoder can extract task-relevant features in a top-down manner, a critical capability for scaling to large images or dynamic VLM tasks. Additionally, the context we use with the multi-zoom patches is an order of magnitude smaller than the ViT context, even for small ($224 \times 224$) images. A more practical context might include multiple gaze locations simultaneously.

## 10. Open Questions

Our experiments show that the multi-zoom patches are easy to handle by the shallow patch tokenizers used in standard ViTs. This is despite the fact that they represent image data in varying scale, from tiny crops to almost the whole image. Additionally, we saw that training a transformer iteratively with an evolving internal state is quite manageable. The main part with open questions is the policy, i.e. learning where to look.

**End-to-End Training of the Whole System.** With this work we use a two stage approach for training our system on image classification. First, we trained the transformer using random selection of multi-zoom patches. Then with a fixed transformer we trained the policy with GRPO (Shao et al., 2024). This approach works because for image classification training the transformer with randomly selected patches is reasonable. For other more complex tasks this might not be the case. As far as we are aware it is an open question how to train the transformer and the policy that uses the transformer's state as observation in a single training loop. While the transformer is being trained, the distribution of the state vectors will be changing. With this the input observations for the policy will be changing as well. On top of that, the training dynamics and the training loop for reinforcement learning algorithms like PPO (Schulman et al., 2017) and GRPO (Shao et al., 2024) are very different from the ones for training a transformer with supervised or self-supervised learning.

**Large Scale Self-Supervised Pretraining.** We strongly believe that powerful and general vision encoders should be pretrained in a self-supervised manner on large amounts of data. The question about end-to-end training of the transformer and policy is still valid. However, there are additional open questions. For example, what is the goal of the policy while pretraining. Let us say we are training a vision model in self-supervised way, e.g. training with the DINO (Caron et al., 2021) objective, but given multiple steps to look at different locations. What is the goal of the policy in this case? How should we compute the rewards for individual actions. Additionally, we want to train a task-driven encoder, but we

are pretraining on a single self-supervised task which will not be encountered after the pretraining stage.

**Should the Policy be Trained with Reinforcement Learning?** While RL is our current choice for driving task-specific behavior, its training dynamics and current performance compared to standard ViTs prompt the question of whether it is the optimal paradigm. Handling this task-driven aspect efficiently is a primary open question we wish to present to the community.

If there is a way to make training the policy differentiable, we might be able to train the system end-to-end. A promising approach here might be based on implicit neural representation of images (Ma et al., 2024; Xie et al., 2022). Beyond this, we identify three alternative pathways to explore. First, *differentiable routing*: using techniques like Gumbel-Softmax to create straight-through estimators could allow the network to learn discrete "where to look" sampling locations via standard backpropagation. Second, *imitation learning*: since human vision operates iteratively much like our proposed framework, human eye-tracking datasets could serve as a strong supervised prior for pre-training a task-driven policy. Finally, *saliency heuristics*: the policy could be bootstrapped using lightweight, traditional saliency or edge-detection filters to guide initial glimpses before a learned policy takes over.

**Large Scale Pretraining Only for the Transformer.** Another option might be to pretrain only the transformer with a fixed or random policy. Then add the policy aspect only when finetuning for particular tasks. Such solution might be in conflict with the task driven property we desire.

## 11. Related work

### 11.1. From Historical Baselines to Spatial Transformers

Prior to the deep learning era, computer vision relied heavily on hand-crafted feature extractors like SIFT (Lowe, 2004) combined with Bag-of-Visual-Words (BoW) representations (Csurka et al., 2004). A key advantage of this historical paradigm was its inherent invariance to global image dimensions; local features were extracted independently of the overall resolution and aggregated into a fixed-size representation. With the advent of deep learning, these flexible designs were largely superseded by Convolutional Neural Networks (CNNs) (LeCun et al., 1998; Krizhevsky et al., 2012; Simonyan & Zisserman, 2014; Szegedy et al., 2015; He et al., 2016), which traded size-agnosticism for vastly superior representation learning by processing spatial features on rigid grids at varying resolutions.

Currently, the most popular transformer architecture is the ViT (Dosovitskiy et al., 2021), a pure backbone that splits

images into fixed-size patches (e.g., $16 \times 16$) treated as tokens. ViT operates at a constant resolution and scales quadratically with pixel count due to global self-attention. While DeiT (Touvron et al., 2021) improved data efficiency, subsequent works heavily modified the architecture to reintroduce hierarchical features, local attention windows, pooling mechanisms, or hybrid CNN designs to improve efficiency (Wang et al., 2021; Liu et al., 2021; 2022; Fan et al., 2021; Heo et al., 2021; Graham et al., 2021; Chen et al., 2021; Chu et al., 2021; Yang et al., 2021).

More recently, the community has pushed towards variable-resolution processing with architectures like FlexiViT (Beyer et al., 2023), NaViT (Dehghani et al., 2023), and ViTAR (Fan et al., 2024). However, while these models achieve greater flexibility regarding input dimensions, they fundamentally remain bottom-up feature extractors. Consequently, they are still prone to distribution shifts when the image size changes drastically—forcing all layers to deal with different levels of detail—and they remain computationally expensive for modern multi-megapixel cameras. This contrasts directly with our proposed top-down, image-size agnostic approach.

### 11.2. Methods Inspired by Fovean Vision

The fovea is a small region of the human retina where light receptor concentration is highest, providing peak resolution in a $1 - 2°$ circle around the gaze direction. Because resolution drops rapidly outside this center, humans rely on constant, highly efficient eye movements (saccades) to sample scene details. This flexible process has long inspired computer vision. (Schmidhuber & Huber, 1991) introduced seminal work on fovean-inspired models using a learned policy and world model. Later, (Mnih et al., 2014) and (Ba et al., 2014) proposed reinforcement learning policies to direct the next glimpse using multi-resolution patches similar to ours. However, their patches are extracted bottom-up rather than top-down, and rely on standard LSTM (Hochreiter & Schmidhuber, 1997) RNNs.

A main goal of our position paper is to revisit these foundational ideas with modern architectural components. Encouragingly, recent literature demonstrates a renewed community interest in active vision, as evidenced by works like SUGARL (Shang & Ryoo, 2023), the active vision framework by (Pardyl et al., 2023), and Mind the GAP (Kolner et al., 2025). Furthermore, contemporary methods like (Lukanov et al., 2021), foveation pooling (Jonnalagadda et al., 2021), and unsupervised top-down attention (Burt et al., 2020) validate the growing need for flexible, saccade-like processing over static, grid-based extraction.

### 11.3. Iterative Transformers with Evolving Internal State

A key component of our proposed method is using a transformer iteratively with a limited context while evolving an internal state. This treats the transformer like an RNN, but without propagating gradients between iterations. Early works explored this by treating internal states as memory: Transformer-XL (Dai et al., 2019) cached hidden states for segment-level recurrence, which the Compressive Transformer (Rae et al., 2019) further compressed. GMAT (Gupta & Berant, 2020) and Memformer (Wu et al., 2020) added dedicated learned memory tokens and dynamic external memories, respectively, though they still relied on long contexts or backpropagation through time over long sequences.

More recently, the Recurrent Memory Transformer (Bulatov et al., 2022) added memory as learnable tokens appended to the input, an approach that has recently been scaled to process up to one million tokens (Bulatov et al., 2023). Additionally, the recent emergence of State Space Models (SSMs) in vision, such as Vision Mamba (Zhu et al., 2024), highlights a strong and growing appetite for recurrent, state-evolving visual architectures. While these contemporary solutions share similarities with our proposal, a crucial difference remains: they typically still rely on expensive backpropagation through time (Robinson & Fallside, 1987; Mozer, 2013; Werbos, 1988), whereas our top-down iterative process avoids this bottleneck entirely.

## 12. Alternative Views

It is our view that having vision encoders to be image size agnostic is self-evidently desirable. Note that this does not mean that the model is completely independent of the image size. Strictly following this might be impossible. The main idea behind this statement is that the computational requirements of a vision encoder should not be increased by increasing the size of the image, unless the task itself gets more difficult with the increase of resolution. Such examples may include camouflaged object detection, or trying to retrieve all the text in an image. Increasing the resolution might make a lot more text available to parse. As such we do not provide alternative view for the desired property of being image size agnostic.

Things are a bit more nuanced with the task driven property. In our opinion the game where the image is hidden before asking questions about it, is a good example why task driven encoders are desirable. However, there is merit in encoders being task agnostic. Modern models like DINOv2 (Oquab et al., 2023) are task-agnostic and perform very well on multitude of tasks. Their strong advantage is that they are very easy to use. One can easily use DINO as a backbone in any system that requires visual perception. In our view task

driven encoders are going to be eventually better and more efficient than task-agnostic ones. However, reaching this point is going to be challenging. Similarly to self-supervised and supervised learning. For a long time it was believed that self-supervised models are better, but their performance did not match that of supervised ones. Today models pretrained in self-supervised manner outperform supervised ones on most of the benchmarks.

A broader philosophical alternative to our perspective might be seen in Richard Sutton's "Bitter Lesson" (Sutton, 2024), which posits that leveraging massive computation ultimately prevails over embedding human-engineered knowledge. Because our fixed-size context and iterative process are inspired by biological foveal vision, our approach might initially appear to contradict this lesson. However, we view our method as aligned with its pragmatic spirit. Computer vision has already heavily leveraged computational scaling through ViTs, and we are now addressing the inherent bottlenecks of scaling those architectures to higher resolutions. Our biological motivation is purely inspirational; the proposed properties – being image size agnostic and task-driven – are fundamentally technical decisions. Rather than artificially mimicking human vision at the expense of compute, our architecture is designed to overcome current scaling limits, ultimately allowing future computation to be leveraged more efficiently.

## 13. Conclusion

In this paper we argued that the future of vision encoders should be focused on models that are image size agnostic and task driven. We also showed a proof-of-concept system that can be used as such an encoder. While this general line of research is not new (Schmidhuber & Huber, 1991; Mnih et al., 2014; Ba et al., 2014). It is our hope to inspire us to revisit these ideas with modern architectural components. We also provided valuable research contributions to this cause – top-down manner of extracting multi-zoom patches in image size agnostic way, and iterative transformer capable of evolving internal state without backpropagation through time.

## Acknowledgements

This research was partially funded by the dAIedge project (HORIZON-CL4-2022-HUMAN-02-02, Grant Agreement Number: 101120726) and the Ministry of Education and Science of Bulgaria (support for INSAIT, part of the Bulgarian National Roadmap for Research Infrastructure). It was also supported with computational resources provided by Google Cloud Platform (GCP).

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

# A. Technical Appendices and Supplementary Material

The supplementary material is organized in the following way. Appendix A.1 provides efficient code in *JAX* to extract multi-zoom patches. Appendix A.2 contains details on how we use GRPO to train a policy where to look. Finally, Appendix A.3 contains implementation details for our *proof-of-concept* solution for image classification.

## A.1. Extracting multi-zoom patches

For completeness we provide efficient code in JAX for extracting the multi-zoom patches in a top-down manner. Since we extract patches of consistent sizes the extraction process can be easily parallelized. See Algorithm 1. The function itself can be used with `jax.vmap` to parallelize the computation for a whole batch of images.

---

**Algorithm 1** Function to extract multi-zoom patches for an image.

---

```
def extract_patches(image, center, patch_size, num_patches, max_z):
    height, width = image.shape[:2]
    # Get the zoom levels
    zs = jnp.linspace(0, max_z, num_patches)

    # Center of each patch in pixels
    center_y = center[1] * height
    center_x = center[0] * width

    def extract_single_patch(z):
        """Extract for a given zoom level."""
        # Patch size in the image space
        patch_img_size = min(width, height) / 2**z
        scale_factor = patch_size / patch_img_size
        translate_x = patch_size / 2 - scale_factor * center_x
        translate_y = patch_size / 2 - scale_factor * center_y
        return jax.image.scale_and_translate(
            image,
            shape=(patch_size, patch_size, 3),
            spatial_dims=(0, 1),
            scale=jnp.array([scale_factor, scale_factor]),
            translation=jnp.array([translate_y, translate_x]),
            method="bilinear",
        )

    # Batch extract for all zoom levels
    return jax.vmap(extract_single_patch)(zs)
```

---

## A.2. Details on learning where to look with GRPO

Once we have trained the transformer and the task head with random gaze locations it is time to teach a policy where to look. We use the group relative policy optimization algorithm (GRPO) (Shao et al., 2024). The internal state from the transformer is the observation state visible to the policy. The rewards comes from how quickly we decrease the cross entropy loss (since we only work with classification here). Given an input image $i$, we use an old version of the policy $\pi_{\theta_{old}}$ to collect $G$ traces $\{o_1, ..., o_G\}$. A trace $o_i$ consists of $n$ actions $o_i = (s_1, a_1, s_2, ..., a_n, s_{n+1})$. We optimize the following objective:

$$\mathcal{L}(\theta) = \mathbb{E}[i \sim P(I), o \sim \pi_{\theta_{old}}(O|i)]$$

$$\frac{1}{G} \sum_{i=1}^{G} \frac{1}{|o_i|} \sum_{t=1}^{|o_i|} \left\{ \min \left[ \frac{\pi_\theta(a_{i,t}|s_{i,t})}{\pi_{\theta_{old}}(a_{i,t}|s_{i,t})} \hat{A}_{i,t}, \text{clip} \left( \frac{\pi_\theta(a_{i,t}|s_{i,t})}{\pi_{\theta_{old}}(a_{i,t}|s_{i,t})} 1 - \epsilon, 1 + \epsilon \right) \hat{A}_{i,t} \right] \right\}. \tag{1}$$

$\hat{A}_{i,t}$ are the group normalized advantages. To understand how they are computed, we first need to define how to get the rewards. We employ two types of rewards. The first one is an end of the episode reward shared across all actions. It is based on the loss after the end of the episode. The second one is an immediate reward for each action defined by the reduction of the task loss. With the first approach the unnormalized advantages $\alpha_{i,t}$ are defined as,

$$\alpha_{i,t} = \log \mathbb{P}\left(\tau(s_{i,n+1}) = y\right). \tag{2}$$

$\tau(s)$ is the output of the task head and $y$ is the ground truth label. In essence, the reward for each action is the negative cross entropy loss, computed at the end of the trace, i.e. with the task head output from the last state $s_{n+1}$.

With the second approach the un-normalized advantage $\alpha_{i,t}$ is defined as,

$$\alpha_{i,t} = \frac{l_{i,t} - l_{i,t+1}}{l_{i,t} + l_{i,t+1}}, \tag{3}$$

where $l_{i,t}$ is the classification loss after applying the task head to state $s_{i,t}$, i.e. $l_{i,t} = -\log \mathbb{P}(\tau(s_{i,t}) = y)$. This is basically the improvement ratio of the loss based on the current action. We have both the current and the next loss in the denominator to keep the reward symmetric around 0. The final group normalize advantages are defined as,

$$\hat{A}_{i,t} = \frac{\alpha_{i,t} - \bar{\alpha}_t}{\hat{\sigma}(\alpha_t)}. \tag{4}$$

Here $\bar{\alpha}_t$ is the mean and $\hat{\sigma}(\alpha_t)$ is the standard deviation of $\alpha_{.,t}$. Note, that we normalize across the traces in the group, but separately for each time step. This is particularly important for the second approach for the advantages using only the immediate reward. The reward during the early steps is typically much higher then that for the latter steps. This is because we start with very low confidence about what is the class of the image, but once it is high it is much harder to increase further. This is why we need to normalize separately for each time step.

Both ways of defining the reward have their pros and cons. With an end of episode reward shared across all actions we attribute the same reward for each action. However, some actions are good and some actions are not good in the same trace. There are specific challenges with image classification. It is a task where a random policy performs very well, and bad actions along the way (looking at uninformative place) does not hurt the performance. This is also the reason we use GRPO instead of Proximal Policy Optimization (PPO) (Schulman et al., 2017). It is challenging to train a good critic when from any state a couple of good actions will lead to a low loss. The second approach for computing the loss is more direct – rewarding actions based on their immediate contribution to decreasing the loss. The challenge here is normalizing the advantages. The starting point for each action is different and thus it is hard to fairly normalize the advantages.

### A.3. Training setup and implemenation details

All our experiments are performed on the ImageNet-1K (Deng et al., 2009) dataset on a TPU v4-32 pod.

#### A.3.1. TRANSFORMER

For the main transformer we use implementation similar to the ViT implementation of DINOv2 (Oquab et al., 2023). The model we use is compatible in size with the ViT-Base models. It consists of 12 layers. The embedding dimension of each token is 768 (spread across 12 heads). The input to the transformer contains $M$ patch tokens and $N$ state query vectors. If it is the first iteration, the $N$ state query vectors are the learned task prompt. In subsequent runs the $N$ state query vectors are the output state from the previous run. Similarly to DETR (Carion et al., 2020) the state query vectors are added to each layer of the transformer as skip connections (He et al., 2016). The $M$ token inputs are the tokenized multi-zoom patches centered at a single location, summed with their respective positional embeddings. The position embeddings are defined on a three dimensional input $(x, y, z)$, where all values lie in the interval $[0-1]$. The positional embeddings are computed with a small MLP network with 3 inputs and 768 outputs. Note that the zoom level $z$ is also a value between 0 and 1. The actual values when extracting the patches are between 0 and $Z_{max}$, but they are scaled to be in the interval $[0-1]$ so that the input to the positional embedding module is normalized. All of the $M$ tokens are centered around a fixed center $(x, y)$.

#### A.3.2. TASK HEAD

Since the only task we use in our experiments is classification we have a simple head that combines the $N$ states vectors through a learned linear combination. Then a simple MLP module is used to output $K$ logits, where $K$ is the number of classes.

#### A.3.3. POLICY

Since we extract multi-zoom patches with the same fixed center, the policy's action can be described only with the $(x, y)$ coordinates. Hence, we use continuous actions, returning a tuple of values between 0 and 1. We use a DETR-like transformer to parametrize the action distribution which is a mixture of $K$ Gaussians. The transformer contains $K + 1$ learned query vectors (one for each mixture and one for the categorical distribution to select which Gaussian to sample from) and does

cross-attention to the $N$ state vectors. Then with a simple head we extract the mean coordinates for each Gaussian of the mixture. The standard deviation is a fixed parameter during training. We use deterministic actions during inference. We opted against representing the action with a single Gaussian, because this assumes there is a single good action from each state. This is clearly not true with the classification task.

### A.3.4. TRAINING STAGES

As mentioned we train the whole system in two distinct stages. In stage 1 we only train the transformer with the task head using a random policy. Each episode contains $8$ steps and in each step we select the center $(x, y)$ uniformly at random. Training is done for 300 epochs with batch size of $1024$. The learning rate is $5 \times 10^{-4}$ decayed with cosine schedule (Loshchilov & Hutter, 2019), after a linear warmup. In this stage we also use MixUp (Zhang et al., 2017) augmentation.

In stage 2, the trained transformer and task head are frozen. And we only train the policy. We do $4$ epochs over the whole ImageNet dataset with batch size $4096$. For each batch we collect $G = 16$ traces of length $8$. This is the data for the inner epochs for GRPO. We optimize the GRPO objective for $8$ inner epochs for each batch.

