# OpenReview forum: "Position: Vision Encoders should be Image Size Agnostic and Task Driven"
_ICML.cc/2026/Position_Paper_Track — ICML 2026 Position Paper Track regular_

### Official Review · Reviewer_5gQx · 2026-02-19

**Significance:** 2
**Argument Clarity:** 2
**Rating:** 4
**Confidence:** 5

**Questions:**

Please refer to the weakness sections above.

Some major issues include:

- In the introduction, the authors discuss a lot of insights from the biological community and bilogical vision. However, no citation is supported these arguments.

- Computer vision tasks are diverse and complicated. Some need local details but some only need high-level semantics. If the authors already acknowledge a next-generation vision encoder need to be task adaptive, ten some need top-down paradigm for global semantics while oter need bottom-up paradigm for local details. Then, is it really a good path to be invariant to either image size or feature map size? Or it is better to also be size adaptive?

- For the system framework in Fig.1, there are multiple flaws and concerns. First, why it needs two tasks and two polices? wyat if we get more than two tasks and policies? Will it support? Will it be scable to multiple tasks and polices? Besides, will it support more states?

- Then, maybe it is more suitable to use recurrent modeling such as RNN and Mamba inside, right? Similar issue in paradigm in Fig.2.

- In Sec.3 and Fig.7, authors already acknowledge the different zoom in can lead to different resoluton, some of them are clear but some are super blurred. Is there some possible path to cope this issue in a size invariant vision encoder? If we need something like super resolution, what is the price?

- In experment section, the performance improvement is marginal. Then, what is computational cost? If it is super high with a marginal improvement, then its meaning is not significant.

- Besides, the experimental outcome lacks explaniablity. Some visualization to explain would be appreciated.

- Related work section is not good. References are mostly 2020-2021. More new references are needed.

Some minor presentation issues are also welcomed to be resolved.

**Alternative Views Section:**

Yes

**Compliance With Llm Reviewing Policy A Conservative:**

Affirmed.

**Discussion Potential:**

3

**Final Justification:**

After reading the rebuttal, I think the major concerns have been addressed. So I improve my score from weak reject to borderline accept.

**Paper Summary:**

This paper raises an important issue in both computer vision and representation learning, that is, how to design the vision encoder?

The authors argue that the vision encoder should be effective enough. Based on this, two improtant properties, namely, image size agnostic and task adaptive, are discussed.

Some arguments and preliminary outcomes are presented, with discussion of related work and alternative view.

**Position:**

Yes

**Position In Title:**

Yes

**Related Work:**

2

**Strengths And Weaknesses:**

**Pros:**

+ How to design the next generation vision encoder is an important question, as existing Transformer paradigm is more suitable for language rather than image.

+ Overall the paper is easy-to-follow and well written.

**Cons:**

- In the introduction, the authors discuss a lot of insights from the biological community and bilogical vision. However, no citation is supported these arguments.

- It is adviced to present a teaser or illustration figure in the introduction to highlight the general idea.

- Most crticially, computer vision tasks are diverse and complicated. Some need local details but some only need high-level semantics. If the authors already acknowledge a next-generation vision encoder need to be task adaptive, ten some need top-down paradigm for global semantics while oter need bottom-up paradigm for local details. Then, is it really a good path to be invariant to either image size or feature map size? Or it is better to also be size adaptive?

- For the system framework in Fig.1, there are multiple flaws and concerns. First, why it needs two tasks and two polices? wyat if we get more than two tasks and policies? Will it support? Will it be scable to multiple tasks and polices? Besides, will it support more states?

- Then, maybe it is more suitable to use recurrent modeling such as RNN and Mamba inside, right? Similar issue in paradigm in Fig.2.

- For subtitles and subcaptions, usage of captial letter is not consistent.

- In Sec.3 and Fig.7, authors already acknowledge the different zoom in can lead to different resoluton, some of them are clear but some are super blurred. Is there some possible path to cope this issue in a size invariant vision encoder? If we need something like super resolution, what is the price?

- In experment section, the performance improvement is marginal. Then, what is computational cost? If it is super high with a marginal improvement, then its meaning is not significant.

- Besides, the experimental outcome lacks explaniablity. Some visualization to explain would be appreciated.

- Related work section is not good. References are mostly 2020-2021. More new references are needed.

**Support:**

2

---

> ### Author Rebuttal · Authors · 2026-03-31
>
> We thank the reviewer for reviewing and acknowledging our work. Here are our answers to the asked questions:
>
> ## Vision Tasks are Diverse and Complicated. Why not Size Adaptive?
> We thank the reviewer for the deep and highly nuanced question. It is a great conversation starter, which shows that our paper “**stimulates constructive, civil discussion**” as intended by the position paper track. We agree that vision tasks are diverse and complicated, and this is why we advocate for a task-driven approach. It might look that there is a contradiction with the image size agnosticism, but with the solution we are proposing this is not the case. We refer the reviewer to Sec 12 (L387-394 right column) about the interpretation of the image size agnostic property. Some tasks like super resolution require the compute to scale with the image size, and this is supported in our proposal. In fact, in work that is an extension of this paper we show that one can train a hybrid model that can equally act as a ViT and an iterative foveal transformer, thus achieving size adaptability for tasks like super resolution.
>
> ## Figure 1, 2 & Architecture Clarifications
> We are afraid that the reviewer might be confused due to misunderstanding of Fig 1. Note that our proposed method is indeed recurrent in nature. **Fig 1 depicts the unrolled version of the recurrent process**. There are no two policies or tasks, instead it is the same policy at action in a sequence of two steps, trying to solve the same task. The policy and task head are the essence of the task-driven behaviour. Fig 2, depicts how we use the transformer as a recurrent model without backpropagation through time. The state acts like a hidden state in an RNN, it is the model’s evolving memory of what it has seen so far. We will explicitly state in the caption of Fig 1 that this is an unrolled version of a recurrent process to avoid this confusion.
>
> ## Why not RNN or Mamba
> Our proposed solution highly resembles an RNN, only without backpropagation through time. Mamba would be an exciting architectural implementation which we leave for future exploration. The concrete architecture here only supports the feasibility of our position.
>
> ## Updating Related Work references
> We will expand Section 11 to reflect more recent literature: (**11.1**) NaViT (Dehghani'23), FlexiViT (Beyer'23), and ViTAR (Fan'24) as variable-resolution ViTs that still rely on bottom-up processing, contrasting with our top-down approach. (**11.2**) Mind the GAP (Kolner'25), Pardyl'23, and SUGARL (Shang'23) showing renewed community interest in active vision. (**11.3**) The 2023 Recurrent Memory Transformer scaling to 1M tokens, and SSMs (Vision Mamba'24) evidencing appetite for recurrent, state-evolving visual architectures.
>
> ## Citations for Biological References
> We agree on the importance of grounding biological inspiration. The camera-ready introduction adds foundational citations: Lennie (2003) for the metabolic cost of selective efficiency, Hubel & Wiesel (1962) for CNNs/visual cortex, Strasburger et al. (2011) for foveal acuity decline, and Yarbus (1967) illustrating that saccades are fundamentally task-driven.
>
> ## Sec 3 & Fig 7 comment (likely Fig 5)
> The reviewer notes the largest multi-zoom patches appear highly blurred. This is an intended feature mimicking human peripheral vision, providing a global context for the policy to decide where to direct its high-resolution "fovea" next. If the task is super resolution, our adaptive policy can simply extract a dense, high-res grid. The computational price then simply equals a standard ViT, seamlessly switching between efficient multi-zoom saccades and dense processing.
>
> ## Experiments & Computational Cost
> We refer the review to our answer to reviewer **1vZj** regarding our experimental results; and to our answer to reviewer **h9rt** regarding computational efficiency.
>
> ## Explainability and minor comments
> To improve explainability, we will include qualitative examples in the final revision.
> Additionally, we will plot the step-wise performance increase (L289-293 right column) to explicitly demonstrate the evolving internal state functioning as memory. Finally, we thank the reviewer for acknowledging our paper is easy to follow and spotting minor comments like missing teaser and inconsistent subtitle capitalization. We will amend this in the final version.

---

> > ### Author Rebuttal · Reviewer_5gQx · 2026-04-04
> >
> > After reading the rebuttal, I think the major concerns have been addressed. So I currently am more positive to this work.

---

### Official Review · Reviewer_D5qZ · 2026-03-08

**Significance:** 3
**Argument Clarity:** 3
**Rating:** 5
**Confidence:** 4

**Questions:**

Please see above Weaknesses.

**Alternative Views Section:**

Yes

**Compliance With Llm Reviewing Policy A Conservative:**

Affirmed.

**Discussion Potential:**

3

**Final Justification:**

My concerns regarding BoW, Pyramid models, and action relying on policy have been resolved.

**Paper Summary:**

This paper argues that vision encoders should be image size agnostic and task driven.
The position comes from a unique perspective: efficiency of biology systems such as human vision to selectively focus via foveation. Then this paper explains the motivation for image size agnostic and task driven with intuitive examples.

The proposed solution involves a Policy to iteratively query the image and process with a Transformer in a recurrent manner, as shown in Figure 1 and 2. Moreover, multi-zoom patches serve as the feature representation with the coordinate center and scale as parameters. Proof-of-concept results demonstrate the feasibility for image classification w.r.t. ViT on ImageNet-1K. Related works and alternative views are included.

**Position:**

Yes

**Position In Title:**

Yes

**Related Work:**

2

**Strengths And Weaknesses:**

### Strengths
- The stated position of this paper is important and reasonable. It finds that existing ViT encoders are trained on fixed patches level (e.g., $16\times 16$), which are less efficient and size dependent. In addition, existing self-supervised ViTs are trained to be general and universal, e.g., DINOv3 is applicable for diverse tasks. Therefore, this paper proposes the position that vision encoders should be image size agnostic and task driven, which will mimic the behavioral treat of biology vision systems w.r.t. efficiency. The stated position addresses existing limitation of Vision encoder in the deep learning and especially ViT era.
- The proposed conceptual solution is complete in both overall design and implementations (Figure 1 and 2). Policy mimics the selective vision foveation in biology system. Recurrent updates without back propagation is practical. A Proofo-of-Concept implementation is conducted on ViT and shows several important feasibility perspectives of the position.
- Figure 3 and 5 are intuitive and illustrative, which can lead to further discussion.
- Besides the stated position, this paper describes an intuitive example in Sec 3: when the task is given, vision models can be more efficient and better performed. This examples is easy to understand.

### Weaknesses
- While deep learning (DL) models for vision suffer from the efficiency and image size dependent issue, the pre-DL methods such as SIFT+BoW solution is less dependent of this issue. In both the Related Work and Alternative Views section, this is not mentioned or discussed. A revisit to historical design can be helpful for future discussion.
- Moreover, the Multi-zoom patches in Figure 4 is more like a revisit to the Pyramid models on top pf SIFT+BoW local features. Revisit and sufficient discussion are beneficial for the community.
- The proposed Action highly relies on Policy to select the focusing position; joint training of ViT and policy network is not easily implemented. While the overall position is important, the action is somewhat less comprehensive.
- Typos.
>  Incorrect use of double quotation symbols.
> gaze(step) lacks a space.
> accumulat-> accumulat2

**Support:**

3

---

> ### Author Rebuttal · Authors · 2026-03-31
>
> We thank the reviewer for their insightful comments and taking the time to review our work. We are also thankful for pointing out minor issues and typos. Here are our responses to the remarks.
>
> ## Connection to SIFT and BoW
> We thank the reviewer for highlighting this important historical context. We agree that the pre deep learning era, specifically methods utilizing SIFT and Bag-of-Visual-Words (BoW), possessed an inherent invariance to image sizes that modern deep learning architectures have largely lost. In our final version, we will dedicate a paragraph in the Related Work section to revisit this historical design. We will frame our position as a step to reclaim the size-agnostic flexibility of the SIFT+BoW era, but equipped with the powerful representation learning capabilities of modern transformers.
>
> ## Regarding Multi-zoom Patches and Pyramid Models
> Our proposed Multi-zoom patches (Figure 4) do indeed share a strong conceptual relation with the classic Spatial Pyramid Matching (SPM) models built on top of local features. Just as SPM extracts features across increasingly fine sub-regions to capture both local details and global spatial layouts, our multi-zoom approach seeks to provide a vision encoder with hierarchical, multi-scale context. We will explicitly cite and discuss Spatial Pyramid models in our revision and their relation to the multi-zoom patches.
>
> ## Action Relies on RL Policy
> Achieving highly performant task-driven policies through RL is one of the main challenges of our proposal. It is also the main open question we like to put to the community. See more about alternatives in our response to **1vZj**.

---

> > ### Author Rebuttal · Reviewer_D5qZ · 2026-04-02
> >
> > My concerns regarding BoW, Pyramid models, and action relying on policy have been resolved.

---

### Official Review · Reviewer_1vZj · 2026-03-08

**Significance:** 2
**Argument Clarity:** 3
**Rating:** 5
**Confidence:** 4

**Questions:**

1. As it is a position paper, it is difficult to ask questions regarding techniques, but besides RL, what other methods could be useful to push the task-agnostic part? Also, are there ideas regarding the framework for tasks like VQA instead of classification?
2. Multi-Instance Learning (MIL) is an active research area where images are so large that it cannot be fully stored in memory. I believe MIL research is heavily aligned with the position paper, would the authors be willing to provide any discussion on this?

**Alternative Views Section:**

Yes

**Compliance With Llm Reviewing Policy A Conservative:**

Affirmed.

**Discussion Potential:**

2

**Final Justification:**

I update my score as the rebuttal addresses my questions

**Paper Summary:**

In this position paper, the authors advocate for vision encoders to be size-agnostic and task-driven. Current vision encoders are transformer-based which are trained/tied to specific image sizes. Even though CNNs are size-agnostic, they process kernel-wise and so patterns can change with image size. Furthermore, regarding task-specific, the authors mention that the task should be known beforehand and be used to guide the vision encoder. Otherwise, the encoder will consider global patterns in the image, possibly discarding some information which would be needed later. Hence, knowing what the task is beforehand can be used to guide the encoder to focus on those specific parts.

**Position:**

Yes

**Position In Title:**

Yes

**Related Work:**

2

**Strengths And Weaknesses:**

**Strengths**
- The authors support the position with good reasoning. They give a parallel between human visual understanding and vision encoders: humans process surroundings at all resolutions, and if given a task, humans focus on exactly those regions. In a similar vein, vision encoders should be able to process input at all image sizes, discard redundant repetitive patches. Furthermore, based on the provided task, the encoders should modify which region of the image to look at.
- The authors provide evidence to support their position with experiments on ImageNet-1K dataset. Their proposed framework has promising results.
- The authors acknowledge hierarchical frameworks in related works, and mention that their position is continue this line of work, and nudge the community into looking into this again. They also discuss recent vision-language models (VLMs) but stress that they models process images first and then process text later. This is a paradigm the authors want to change.
- They also provide alternative view section where they discuss the merits of task-agnostic methods (like DINO) which have been successful as well.

**Weaknesses**
- I understand it is a position paper, but the next steps to advance 'task-driven' position is still a bit unclear. The learned policy's performance is much lesser than ViT's, and we do not know if RL is even the right paradigm.
- The authors do not discuss Multi-Instance Learning (MIL) techniques which is one area of active research that is heavily aligned with the paper's position.

**Support:**

3

---

> ### Author Rebuttal · Authors · 2026-03-31
>
> We thank the reviewer for spending time to review and acknowledge our work. Here are our responses.
>
> ## Experimental Performance (cross response to 5gQx)
> The reviewer rightly notes that the experimental results in our work are not as good as ViTs.
> As a position paper, our experiments were purposefully kept to a minimum, prioritizing only the validation of our approach's feasibility rather than achieving state-of-the-art metrics. In ongoing work extending from this paper, we are already able to match ViT’s performance at a fraction of the computational cost when processing large input images. In this position paper our goal is to show that the technical direction we propose is possible, and our experiments achieve this.
>
> ## Advancing the "Task-driven" Aspect & Alternatives to RL (cross response to D5qZ)
> Thank you for highlighting this. Handling the task-driven aspect efficiently is indeed the primary open question we wish to present to the community. While RL is our current best bet for driving this behavior (as briefly discussed starting at L366 left column), the reviewer rightly asks what other methods could be explored. Alternatives to RL could include:
> * **Differentiable Routing**: Using techniques like Gumbel-Softmax to create differentiable, straight-through estimators that allow the network to learn discrete "where to look" sampling locations via standard backpropagation.
> * **Imitation Learning**: Pre-training the policy network using human eye-tracking datasets. Since human vision operates exactly as our proposed framework envisions, human gaze data could serve as a strong supervised prior for a task-driven policy.
> * **Saliency Heuristics**: Bootstrapping the policy using lightweight, traditional saliency or edge-detection filters to guide the initial glimpses before a learned policy takes over.
>
> We will add a dedicated paragraph discussing these alternative pathways for the task-driven policy.
>
> ## Framework Ideas for VQA
> We briefly touch upon Vision Language Models (VLMs) in Sec 5. While not part of our experiments, our proposal holds the key to an important paradigm shift for VLMs and VQA. Currently, VLMs typically extract static features from the image and then pass them to an LLM alongside text. Under our framework, an iterative, flexible vision encoder would incorporate the **text prompt during the encoding process to actively guide which features to extract from the image**. For instance, the question "What is the brand of the car?" would act as the "task", driving the encoder's policy to focus specifically on the car to extract high-resolution features, while ignoring the background. Classification is used as a proof-of-concept; we are actively looking into incorporating multi-modal tasks into extensions of this work.
>
> ## Multi-Instance Learning (MIL)
> We highly appreciate the reviewer drawing this excellent connection. MIL is highly relevant, particularly for large-scale images (e.g., gigapixel pathology slides or satellite imagery) that cannot fit into memory.
>
> Our proposed framework offers a top-down approach, which can be complimentary to bottom-up MIL. Instead of exhaustively processing all instances, an active policy must decide not only where to look, but at what zoom level. Handling this effectively requires reasoning through both visual clues and prior knowledge. For example, when searching a massive satellite image for a tiny object like a lighthouse, a highly compressed, coarse view of the image will lack the necessary resolution to spot it. The policy must learn to use prior context – scanning the coastlines rather than inland – to zoom in and sample the relevant high-resolution instances. We will explicitly include a discussion connecting our top-down position to the challenges of MIL in the final version.

---

> > ### Author Rebuttal · Reviewer_1vZj · 2026-04-03
> >
> > The authors address my questions in the rebuttal

---

### Official Review · Reviewer_h9rt · 2026-03-16

**Significance:** 3
**Argument Clarity:** 3
**Rating:** 5
**Confidence:** 4

**Questions:**

Why is avoiding back-propagation through time important to the position the paper takes?

The lens in human eyes is for focusing, not zooming.  Can you draw a more clear connection to biology for the image size agnostic aspect of the proposed research direction?

**Alternative Views Section:**

Yes

**Compliance With Llm Reviewing Policy A Conservative:**

Affirmed.

**Discussion Potential:**

3

**Final Justification:**

I think the paper should be accepted.  Still not sure I buy the argument about need for scale invariance being a consequence of the density of photoreceptors in the fovea.

**Paper Summary:**

This paper argues research on vision encoders should revisit the saccade mechanism in biological vision systems to enable efficiency and argues a fundamental missing piece in current systems is the lack of task awareness and ability to zoom.  The paper presents a prototype system coupling a recursive transformer model for understanding patches with a reinforcement learning controller for directing where to sample in a larger image.  There are measurements showing that across iterations classification accuracy on ImageNet improves from a low value towards a baseline.

**Position:**

Yes

**Position In Title:**

Yes

**Related Work:**

3

**Strengths And Weaknesses:**

Strengths:
- Improving efficiency is very important and the paper points researchers to reconsider biological existence proof that such efficiency can be obtained and connects the task driven aspect well.
- The idea of making vision task driven is interesting.
- The vision problem solving example was motivating.
- Proto-type system.

Weaknesses:
-  Limited novelty: As the paper does a good job pointing, there have been numerous works that draw inspiration from saccades and the idea of iterating over transformers has been explored before.
- The connection of image size agnosticism to biological examples was not as clear as that for task driven:  Much as one might like to have built in telephoto lenses in my eyes, there is a reason binoculars/telescopes/microscopes were invented.
- The motivation is efficiency, but there is no discussion of the impact on efficiency after the prototype is introduced (even if just to acknowledge that efficiency is not gained).
- The "Alternative Views Section" should address the 'bitter lesson' essay from Sutton, which seems related.

**Support:**

3

---

> ### Author Rebuttal · Authors · 2026-03-31
>
> We thank the reviewer for reviewing our work and for their insightful comments. Here are our responses to the raised remarks.
>
> ## Limited Novelty
> While the ideas behind foveal inspired vision models are not new, we do provide novelty in a few areas. To the best of our knowledge our proposal to use top-down patch extraction for transformers with varying sizes is novel. It also addresses issues with the current bottom-up approach beyond the position in the paper (distribution shifts from changing image resolution). Additionally, while sequential use of a transformer is not new, we are not aware of other works using a transformer like RNN without backpropagation through time.
>
> ## Connection of Image Size Agnosticism to Biological Examples
> The main biological example for image size agnosticism is the combination of foveal and peripheral vision, where we see in high definition only a small area around the center of the gaze, while we have a large field-of-view with low resolution peripheral vision. Understanding the scene consists of rapid gaze fixations called saccades. In terms of computer vision encoders this means that the computational complexity of each step is independent of the image size.
>
> If the resolution of the image is higher than the resolution supported by the fovea, it would be beneficial to use zoom in order to see detail beyond what the multi-zoom patch with highest resolution supports. If we increase the zoom-level for all multi-zoom patches this would be akin to using a “virtual telescope”. This is true for humans as well, if we want to see details on a distant object we have to use binoculars/telescope.
>
> ## Efficiency Discussion (cross-response to 5gQx)
> We thank the reviewer for this question. Computational efficiency is an important aspect of our approach. In the paper we have kept experimental evaluation to a minimum in order to make the arguments for our position the main theme. This is why we have not included charts showing this. However, our solution already exhibits significant efficiency improvements over ViT when images become large. We will include a chart showing this. Unfortunately, we cannot share a chart in the rebuttal format. It will show image size (height or width) on the x-axis and inference time and GFLOPS on the y-axes. While ViT scales quadratically with the number of patches and quartic with the side length of the images, our method remains near constant per step. Running our model with a policy and multi-zoom patches for 8 steps is already an order of magnitude more efficient than ViT when the image size approaches 1000x1000 pixels.
>
> ## Bitter Lesson in Alternative Views
> We thank the reviewer for this excellent remark. Indeed the bitter lesson essay from Richard Sutton is a fantastic example for the “Alternative Views” section. The field of computer vision has already been through the phases from the essay: starting from manual approaches trying to extract human knowledge, to convolutional neural networks, and finally ViTs where the main focus is computational scaling. We are currently past this point and looking for ways to address the remaining bottlenecks. Note, that the biological motivation for our work is purely inspirational. The technical decisions we follow in the image size agnostic and task-driven behavior are pragmatic in nature. Our goal is not to strictly mimic biological foveal vision.
>
> ## Why Avoiding Backpropagation-through-time
> This is not directly related to the position of the paper, but a technical experiment. Training recurrent networks on a large horizon has been historically challenging. Having the ability to train this model without backpropagation through time shows a flexibility on how this can be used in the future for tasks requiring a large number of saccades.
>
> ## The Human Lens is for Focusing, not Zooming
> While this is true, the process of foveation and saccades to multiple gaze locations is about building global image understanding, through local high-resolution (fovea) and global low-resolution (peripheral) data. **Note that the proposed multi-zoom patches are not attempting to simulate zooming, but to take advantage of how the biological foveal vision works** (high resolution center and low resolution periphery). Zooming-in in this situation would be done through offsetting the zoom level of each patch. This would be akin to using a “virtual telescope”.

---

> > ### Author Rebuttal · Reviewer_h9rt · 2026-04-01
> >
> > As per my prior comment: Wouldn't a more plausible biological inspiration for image size agnosticism be that animals are generally mobile and can thus move towards or away from an object of interest?

---

> > > ### Author Response · Authors · 2026-04-03
> > >
> > > We thank the reviewer for this interesting observation. We agree that in biology, physical mobility allows an animal to dynamically control the scale of an object in its visual field by moving closer or further away. At the same time, our biological inspiration focuses on the internal processing mechanisms of the visual system (foveal vs. peripheral) rather than the physical movement of the agent. This physical mobility is an excellent biological inspiration for Embodied AI, where our approach could be useful. When handling static visual inputs, zooming abilities can be seen as a compensation for lack of mobility.

---

### Decision · Program_Chairs · 2026-04-30

**Decision:**

Accept (regular)

**Comment:**

This paper presents a compelling position that future vision encoders must be image size agnostic and task-driven. Inspired by biological foveal vision and saccades, the authors propose a top-down, iterative prototype combining a recursive transformer with a reinforcement learning policy.

Reviewers commended the submission for tackling a crucial efficiency limitation in current Vision Transformers. Reviewer D5qZ praised the intuitive examples and complete conceptual design, while Reviewer 1vZj highlighted the promising proof-of-concept results on ImageNet-1K. The primary weaknesses identified during the review phase included the omission of historical context such as SIFT and Spatial Pyramids, missing biological citations raised by Reviewer 5gQx, and questions regarding the scalability of the RL policy.

The authors successfully addressed these concerns during the rebuttal by integrating discussions on Multi-Instance Learning, historical baselines, and alternative policy training methods. Ultimately, this well-written paper stimulates valuable discussion on the future of efficient representation learning and merits acceptance.